# Cardiopulmonary and Neurologic Dysfunctions in Fibrodysplasia Ossificans Progressiva

**DOI:** 10.3390/biomedicines9020155

**Published:** 2021-02-05

**Authors:** Fatima Khan, Xiaobing Yu, Edward C. Hsiao

**Affiliations:** 1Department of Medical Sciences, Frank H. Netter MD School of Medicine at Quinnipiac University, North Haven, CT 06518, USA; Fatima.Khan@quinnipiac.edu; 2Department of Anesthesia and Perioperative Care, University of California San Francisco, San Francisco, CA 94143, USA; Xiaobing.Yu@ucsf.edu; 3Department of Medicine, Division of Endocrinology and Metabolism, the Institute for Human Genetics, and the Program in Craniofacial Biology, University of California, San Francisco, CA 94143, USA

**Keywords:** Fibrodysplasia Ossificans Progressiva (FOP), cardiac conduction abnormalities, ACVR1, neuropathic pain, cardiac dysfunction, neurological dysfunction

## Abstract

Fibrodysplasia Ossificans Progressiva (FOP) is an ultra-rare but debilitating disorder characterized by spontaneous, progressive, and irreversible heterotopic ossifications (HO) at extraskeletal sites. FOP is caused by gain-of-function mutations in the Activin receptor Ia/Activin-like kinase 2 gene (*Acvr1/Alk2*), with increased receptor sensitivity to bone morphogenetic proteins (BMPs) and a neoceptor response to Activin A. There is extensive literature on the skeletal phenotypes in FOP, but a much more limited understanding of non-skeletal manifestations of this disease. Emerging evidence reveals important cardiopulmonary and neurologic dysfunctions in FOP including thoracic insufficiency syndrome, pulmonary hypertension, conduction abnormalities, neuropathic pain, and demyelination of the central nervous system (CNS). Here, we review the recent research and discuss unanswered questions regarding the cardiopulmonary and neurologic phenotypes in FOP.

## 1. Introduction

Fibrodysplasia Ossificans Progressiva (FOP) is a rare, debilitating disorder characterized by ectopic calcification of soft tissues by heterotopic endochondral ossification (HO) at extraskeletal sites [1]. FOP arises from gain-of-function point mutations in the *Acvr1*/*Alk2* gene, which encodes for the bone morphogenetic protein (BMP) receptor activin receptor Ia/activin-like kinase 2. Phosphorylated by a Type II receptor [2], the ACVR1 receptor with de-novo germline mutations has increased sensitivity to BMP activation [3] to trigger downstream SMAD1/5/8 signaling [4]. Somewhat surprisingly, the abnormal ACVR1 receptor becomes activated by Activin A, normally a competitive antagonist of the wildtype ACVR1 receptor [5,6].

While disease presentations vary, early hallmarks seen in patients with FOP are malformed great toes at birth and migratory soft tissue swellings that are associated with HO formation [1]. The HO typically follows a predictable anatomic pattern, first involving the upper back and neck, then progressing to the trunk, extremities, and scalp as the patient ages [7]. Patients with FOP also have characteristic skeletal malformations including clinodactyly, vertebral abnormalities, shortened and broadened femoral necks, and tibial osteochondromata (Table 1) as well as other findings [8,9,10,11,12,13,14,15,16]. As heterotopic bone continues to form in injured muscles, ligaments, and tendons, functional activities such as breathing, eating, and walking become compromised for patients [7,17,18,19].

Although the presence of malformed great toes and evidence of heterotopic ossification in muscles, tendons, ligaments, fascia, and aponeuroses are sufficient to establish a clinical diagnosis of FOP [20], definitive diagnosis is through genetic analysis of the *Acvr1/Alk2* gene. The classical c.617 G > A (R206H) mutation occurs in more than 95% of cases [3,9]. Sadly, approximately 90% of FOP patients have some form of initial misdiagnosis. The soft tissue swellings in FOP are often mistaken for lymphedema, soft tissue sarcoma, or aggressive juvenile fibromatosis [21]. More than half of patients underwent unnecessary invasive procedures such as biopsies that could trigger robust ossification and result in permanent harm [21].

Inflammation is an important component of the FOP phenotype and most evident through the pre-osseous inflammatory swellings, which form prior to many cases of HO formation [1,19]. Patients with FOP also appear to have a primed immune system with increased levels of inflammatory cytokines released by immune cells through the activated NF-κB pathway [22]. Activin A and BMPs can both regulate the immune system [23,24]. Histopathology samples in early FOP lesions show abundant macrophages, lymphocytes, and mast cells [7,25,26]. Macrophage and lymphocyte-associated myopathy and inflammatory flare-ups have been reported after viral infections [27]. In addition, strategies for depleting macrophages can mitigate injury-induced HO in animal models, suggesting an important contribution of macrophages to FOP pathogenesis [28]. The microenvironment of injured tissue in pre-osseous swelling also contributes to increased BMP pathway signaling. Hypoxic conditions, observed in inflamed, traumatized tissue, often have higher levels of hypoxia inducible factor 1 (HIF1-α), which promotes the retention of the ACVR1 receptor and activates BMP pathway signaling [29].

In addition to the remarkable skeletal abnormalities and HO, various non-musculoskeletal phenotypes have been observed in patients with FOP (Figure 1). These include molecular data suggesting increased sensitivity of the immune system to inflammatory triggers, resulting in inflammatory pre-osseous flare-ups; an increased risk of renal stones; gastrointestinal symptoms; thinning of the skin, hair, and eyebrows; dental abnormalities; and hearing loss [9,12,22,30,31,32,33]. However, these findings are neither well understood nor studied in various genetic mouse models including conditionally inducible *Acvr1^R206H^* lines that have been developed to mimic different aspects of FOP [5,34,35,36,37,38]. This review article focuses on the emerging evidence about cardiopulmonary and neurologic phenotypes in FOP.

## 2. Cardiopulmonary Dysfunction in Fibrodysplasia Ossificans Progressiva (FOP)

### 2.1. Pulmonary Complications in Fibrodysplasia Ossificans Progressiva (FOP)

Patients with FOP often die early due to cardiopulmonary complications from thoracic insufficiency syndrome or pneumonia [39]. A retrospective medical record review published in 2010 of forty-eight deceased patients with FOP by Kaplan et al. found that the median patient age at the time of death was forty years. Kaplan et al. postulated that the development of thoracic insufficiency syndrome in FOP originated from bony chest wall deformities including costovertebral malformations, ankylosis of costovertebral joints, and asymmetric ossification of the back and chest wall, leading to restrictive lung disease, pulmonary hypertension, and right-sided congestive heart failure, eventually causing death [39]. In addition, an autopsy series showed that patients with FOP could develop restrictive pulmonary physiology (likely due to the fusion of the thorax), reduced lung volumes, and changes in right ventricular structure, consistent with thoracic insufficiency syndrome [40]. While hypercarbia from hypoventilation may be the main mechanism causing pulmonary hypertension, it is also hypothesized that dysregulated BMP signaling due to the *Acvr1*/*Alk2* mutation may directly contribute to the development of pulmonary hypertension. Of note, BMP4 is upregulated in hypoxia-induced pulmonary hypertension and promotes pulmonary vascular remodeling [41]. BMP2 is known to promote osteogenesis and chondrogenesis of mesenchymal progenitor cells [42]. In half of patients with chronic pulmonary hypertension, histological evidence of dystrophic calcification in pulmonary vasculature has been observed [43]. Furthermore, mutations in the BMP receptor II gene (*Bmpr2*) are the single most common causal factor for hereditary pulmonary arterial hypertension [44,45,46]. As a type II receptor, BMPR2 forms a complex with ACVR1 to initiate intracellular activation [2]. Thus, overactive BMP signaling through the mutant ACVR1 receptor may contribute to the high incidence of pulmonary hypertension in patients with FOP [39].

### 2.2. ACVR1 and Cardiac Development

The *Acvr1*/*Alk2* gene is essential in BMP-driven cardiac development including morphogenesis and valve formation [47,48]. ACVR1 is abundantly expressed in the endothelial cells of the endocardial cushions, outflow tract, and ventricle of mice [49,50]. ACVR1 is required for the induction of endothelial-to-mesenchymal trans-differentiation during endocardial cushion formation, as observed in mice [49]. ACVR1-mediated BMP signaling is also important for the second heart field and arterial trunk development, myocardial differentiation, right ventricular growth, interventricular, outflow tract, and aortico-pulmonary septation [51]. Deficiencies in ACVR1 are associated with congenital cardiac malformations such as atrioventricular septal defects in knockout mice. Mice that lack expression of the *Acvr1*/*Alk2* gene develop dysregulated myocardial differentiation and development of aortic valve pathologies including bicuspid aortic valve [51,52,53]. A complex interplay between various BMP receptors promotes and inhibits differentiation of the second heart field. The arterial pole of the heart develops from the second heart field, which is mesodermal tissue that gives rise to the myocardium of the interventricular septum, right ventricle, myocardium, and smooth muscle of the arterial trunk [51]. Additionally, upregulated BMP signaling plays a role in the development of angiotensin-II mediated cardiac hypertrophy and fibrosis, as observed in rat cardiac myocytes [54]. Injured tissue in non-FOP patients such as in heart failure can exhibit increased levels of Activin A expression [55]. These normal responses may in turn lead to abnormal signaling in patients with FOP. However, variability in the expression of the *Acvr1*/*Alk2* gene and the normal roles of the ACVR1/BMP pathways in these tissues may contribute to an organ-specific clinical presentation. Taken together, it is clear that ACVR1 and BMP signaling are important mediators of cardiac development and pathology, as demonstrated in cell and animal models. However, whether increased ACVR1 signaling in FOP has an effect on cardiac clinical presentation in patients with FOP needs further investigation.

The role of ACVR1 in cardiac disease has also recently been of interest. A genetic sequencing study of non-FOP human patients with congenital heart defects (CHD) identified variants in the ACVR1 gene as well as other candidate genes [56,57]. Smith et al. characterized two novel variants in ACVR1 occurring in separate individuals with CHD. The L343P variant exhibited dominant negative interference activity in endocardial cushion development in zebrafish embryo experiments and decreased ACVR1 signaling and kinase activity in in vitro studies. The group also reported an additional ACVR1 variant (H286D) along with variants in two other genes normally involved in cardiac development in a patient with Down Syndrome and CHD [57]. In vivo and in vitro studies of this ACVR1 variant demonstrated mild dominant-interfering activity on BMP signaling and reduced BMP signaling capacity, the opposite of what is observed in FOP. The two other genetic variants identified were in ALK3 (G414K) and the epidermal growth factor receptor gene ErbB3 (T116I). Since the patient’s immediate family members who carried combinations of the ACVR1/ALK3, and ALK3/ErbB3 variants did not have CHD, the group concluded that the individual variants were either not penetrant or those combinations of variants were insufficient to cause CHD. The group also concluded that ACVR1 may be a susceptibility gene for CHD when it is combined with variants in other genes or in the presence of trisomy 21. Thus, although ACVR1 is an important player in cardiac development, further clarification of its interactions with other cardiac developmental genes is needed.

### 2.3. Cardiac Abnormalities in FOP

Only a few studies have investigated potential cardiac phenotypes in patients with FOP. Although right-sided congestive heart failure has been reported in patients with FOP, this is currently thought to be secondary to thoracic insufficiency syndrome [39]. In a study by Kussmaul et al. twenty-five patients from the ages of 5–55 were examined by history, physical exam, pulmonary function tests, electrocardiogram, and echocardiogram [58]. There was a high incidence of patients with severely limited chest expansion and low lung volumes, although most had normal pulmonary function and electrocardiograms. Ten patients had evidence of right ventricular dysfunction on electrocardiogram. Compared to the other patients in the study without cardiac dysfunction, these ten patients were older with a longer duration of disease, elevated hemoglobin, and more severe pulmonary function impairment. The authors believed that restrictive chest wall disease was responsible for the right ventricular dysfunction.

Structural cardiac defects have been observed in patients with FOP, but the exact prevalence is unknown. For example, in one case report, the autopsy of a 36-year-old female with FOP but no prior cardiac history who died due to a gastrointestinal disease revealed thickened aortic and mitral valve cusps and short, thickened chordae tendinae of the mitral valve [59]. In another case, a newborn with FOP was found to have unusual ventricular septal hypertrophy on electrocardiogram and echocardiogram [60]. Common causes of ventricular septal hypertrophy in the newborn were ruled out such as maternal diabetes, hypertension, left ventricular outflow tract obstruction, and premature ductus arteriosus constriction due to non-steroidal anti-inflammatories. The authors of this case report hypothesized that the ventricular septal hypertrophy was due to mechanical stimuli from hemodynamic changes at birth. However, given that the patient was known to have FOP, it is possible that aberrant ACVR1 signaling could have led to the observed congenital cardiac changes. Finally, although autopsies on patients with FOP have identified some right ventricular changes, it remains unclear if cardiac structural changes are related to the abnormal physiology in thoracic insufficiency versus a congenital phenotype directly caused by ACVR1 overactivity [40].

The prevalence of other cardiac complications such as cardiac arrhythmias in patients with FOP has also been examined. The FOP Natural History Study (NHS) is a three-year longitudinal, multicenter natural history study that observed 114 patients between the ages of four to fifty-six years old with genetically confirmed FOP. Based on data from baseline and 12-month follow-up assessments, 45.3% of patients presented with abnormal baseline electrocardiograms, with most abnormalities classified as non-specific intraventricular conduction delay observed across all age groups [61]. These changes were unlikely to be of major clinical significance. Other observed conduction abnormalities included right bundle branch block, incomplete right bundle branch block, and first-degree atrioventricular nodal (AV) block. The highest incidence of right bundle branch block was observed in patients from 9–16 years old, while first degree AV block was observed most in patients ages 25–56 years old. Additionally, the incidence of conduction abnormalities in patients greater than 18 years old was higher when compared to a normal, healthy population of Phase 1 clinical trial participants. The conduction abnormalities detected on the electrocardiogram did not correlate with scoliosis, decreased pulmonary function, chest wall HO, or FOP disease severity. Although these conduction abnormalities detected on electrocardiograms were largely considered to not have clinical significance, some patients with FOP who also had echocardiograms showed structural abnormalities. These structural abnormalities were not correlated with electrocardiogram changes.

The clinical significance of the observed conduction abnormalities in patients with FOP remains unknown, as cardiac complications are not thought to be a major part of the FOP phenotype except for the risks associated with thoracic insufficiency syndrome. However, these baseline findings are important since patients with FOP may be found to have cardiac changes during routine medical care. Furthermore, cardiac complications are a major reason for drugs to fail in clinical trials or receive approval [62]. Currently, there are no clinical recommendations for electrocardiogram or echocardiogram monitoring in patients with FOP. However, there is a need for further investigation of the cardiac phenotype in FOP and whether it plays a clinically important role in patient care.

## 3. Neurologic Dysfunction in FOP

### 3.1. Incidence and Prevalence of Neurologic Symptoms in FOP

In addition to the cardiac phenotype, neurologic changes are another major non-skeletal aspect of FOP that remains inadequately examined. There is currently only one published study that has examined the prevalence of neurological symptoms among patients with FOP [63]. The group surveyed 168 patients with FOP from thirty different countries ranging from the ages of 1.5 to 68 years old with a questionnaire regarding neurologic symptoms. The survey study revealed that 51% of patients had at least one chronic neurologic symptom. Symptoms with the highest incidence in the study population were recurrent severe headaches (26.2%), followed by neuropathic pain (10.1%), and other sensory abnormalities (36.3%). However, only the incidence of neuropathic pain was statistically significantly increased compared to the general population [63]. Notably, females with FOP had a greater prevalence of sensory neurologic symptoms compared to their male counterparts (sixteen of the seventeen patients who reported neuropathic pain were female) [63]. Female patients also reported more sensory symptoms such as numbness, tingling, and abnormal heat and cold sensation. In contrast, no difference between sexes was observed for the incidence of non-sensory neurologic symptoms such as seizures, myoclonus, and brain and spinal cord injury [63]. The findings are consistent with other chronic pain conditions that also show a marked female predominance [64,65,66] and sexual dimorphism in the cellular mechanism of neuropathic pain [67].

Interestingly, an increasing prevalence of neurologic symptoms among females with FOP was noted with each decade of life, while an inverse trend was observed among males [63]. Among the 49 menstruating women in the study, 33% (16 patients) reported a worsening of symptoms such as severe headaches and seizures during their menstrual period. The authors also compared the prevalence of neurological symptoms among females younger than 12 years old to those older than 12 years old to assess whether puberty contributed to symptom onset. They found a greater prevalence of neurological symptoms in subjects older than 12 years old. However, there was no statistically significant difference detected in neurological symptom prevalence before and after the age of puberty among males [63].

### 3.2. ACVR1 and Neurological Development

ACVR1 plays an essential role in embryonic development as mice with a complete deletion of *Acvr1* are embryonically lethal [68]. In addition to the impaired chondrogenesis and subsequent craniofacial and cervical vertebral developmental defects in *Acvr1*^−/−^ mice [69,70], BMP signaling is essential for the formation and patterning of the central nervous system (CNS) [71,72,73,74,75,76]. Defective expression of physiological BMP antagonists such as *Chordin* and *Noggin* disrupted the development of the rostral neural ectoderm and downstream malformations in the forebrain on a spectrum including cyclopia, holoprosencephaly, and truncated rostral brain [77]. Lack of *Fortilin*, another physiologic BMP antagonist, resulted in overactivation of the BMP pathway and embryonic lethality in *Fortilin^−/−^* mice [78]. Activin A, a normally competitive wild-type ACVR1 receptor antagonist, is also observed to be significantly elevated in pathological conditions of nervous system injury such as trauma, hypoxic-ischemic brain injury, stroke, and epilepsy, and has shown both neurotrophic and neuroprotective features [79,80,81,82,83,84]. For example, in animal models of excitotoxic and hypoxic-ischemic brain injury, addition of Activin A promoted survival of neurons [81,85,86,87]. Conversely, inhibition of activin signaling counters this neuroprotective effect [80,85]. Interestingly, ACVR1^R206H^ does not require its ligand binding domain to over-activate BMP signaling in zebrafish embryonic dorsoventral patterning [88].

Nakashima et al. demonstrated that exposure of CNS neural precursor cells to BMP2 irreversibly changes their fate to astrocytic differentiation via BMP2 interactions with neurogenic transcription factors [89]. In contrast, BMP2 induces neuronal differentiation by triggering Mash1 transcription factor activity in neural crest stem cells [90,91,92]. Importantly, inhibition of BMP/ACVR1 signaling has been used to specify patient-derived induced pluripotent stem cells (iPSC) into neurons [93,94] to model human neurological disease conditions and identify new therapeutic targets [95,96]. In addition to regulating neural stem cell fate in the CNS, BMPs seem to be able to inhibit myelin production by oligodendrocytes and may represent a therapeutic target for multiple sclerosis and other inflammatory demyelinating conditions [97,98,99]. Furthermore, mice with neuron-specific enolase (NSE) driven expression of BMP4, used to model BMP activity in FOP, and autopsies of patients with FOP, have shown demyelinating CNS lesions [100]. Thus, BMP/ACVR1 signaling has important roles in the nervous system development and studying dysregulated ACVR1 signaling can improve our understanding of the neurological phenotypes observed in FOP.

### 3.3. Neuropathic Pain in FOP

In addition to inflammatory pain experienced during HO flares, a recent analysis of the international FOP Registry revealed 30% to 55% of patients experienced pain during non-flare-up states [101]. In a parallel survey study, patients with FOP reported a higher incidence of neuropathic pain compared to the general population [63]. Most patients with neuropathic pain also reported an additional sensory symptom such as allodynia, numbness, and tingling, commonly in the legs and feet [63].

As anti-inflammatory treatments generally provide suboptimal pain relief in FOP patients, inflammatory mediators such as prostaglandin E2 are likely not the primary contributors to pain hypersensitivity [102,103]. Several ligands of BMP signaling such as the BMPs, TGF-βs, and Activins have been implicated in peripheral nociception in animals. In a study by Follansbee et al., the role of BMP signaling in pain perception was highlighted when they found that RNAi silencing of a member of the BMP signaling pathway decreased ultraviolet injury-induced sensitization in nociceptive neurons in a *Drosophila melanogaster* model. The group further demonstrated that overexpression of the BMP signaling pathway in nociceptive neurons induced thermal hypersensitivity [104]. Interestingly, in the cultured rodent dorsal root ganglion (DRG) neurons, Activin A increased expression of nociceptor neuropeptide calcitonin gene-related peptide (CGRP) [105], a prominent clinical target for pain conditions including migraine [106]. Injection of recombinant Activin A into the paw skin can lead to mechanical allodynia in rats [107] and potentiate capsaicin-induced ionic currents through the transient receptor potential vanilloid 1 (TRPV1) cation channel [108]. Infusion of TGF-β into rats resulted in sensory neuron hyperexcitability and increased nociception, in part via a direct effect on primary sensory neurons mediated by intra-neuronal SMAD3 activation [109]. Additionally, intrathecal administration of TGF-β1 was reported to reduce injury-induced mouse neuropathic hypersensitivity [110]. These findings suggest a potential connection between BMP signaling and hypersensitivity to mechanical and thermal stimuli reported among patients with FOP [63]. This is an area of great interest and active investigation.

### 3.4. Focal CNS Demyelination and Other Neurological Abnormalities in FOP

BMP signaling is essential for the formation and patterning of the central nervous system (CNS) [75]. In transgenic mice overexpressing BMP4 under the promoter of neuron-specific enolase (NSE), decreased conversion of neural progenitor cells into mature oligodendrocytes was observed in the CNS [37]. NSE-BMP4 mice also showed evidence of acquired CNS focal demyelination. In a study by Kan et al. NSE-BMP4 mice showed asymmetric hyperintense lesions of demyelination on T2-weighted magnetic resonance imaging (MRI) imaging of the brain and spinal cord, with lesions correlating histologically on the immunohistochemistry stain [100]. They also found that knock-in mice heterozygous for the ACVR1^R206H^ had consistent lesions of demyelination in the cerebellum, spinal cord, and other regions of the brain. Finally, the group analyzed the MRI images of four patients with FOP, which revealed hyperintense lesions of focal demyelination. One of those patients was a seventeen-year-old female presenting with refractory propriospinal myoclonus. She had evidence of lesions in the left frontal periventricular white matter and the T2–T3 regions of her spinal cord on MRI. MRI evidence of CNS demyelination observed in patients with FOP and in ACVR1^R206H+/-^ mouse model of FOP suggest that aberrant ACVR1 signaling potentially contributes to these focal lesions in the CNS.

In addition to focal CNS demyelination, other atypical neurological findings observed in patients with FOP include myoclonus, structural malformations of the cerebellum and corpus callosum, craniopharyngioma, and brainstem lesions. In a survey study conducted among 407 International FOP Association members, the incidence of myoclonus was found to be higher than observed in the general population [63,100]. Additionally, in a paper by Kaplan et al., a child with one of the most severe manifestations of FOP was found to have agenesis of the corpus callosum and hypoplasia of the cerebellum [9]. Craniopharyngiomas and mild cognitive impairment were also observed [9]. A case report of a three-year-old boy with FOP described a hyperintense lesion of the dentate nucleus in the cerebellum on T2-weighted MRI [111]. Brainstem masses have also been noted in patients with FOP. Recently, in an autopsy of a 75 year old male with FOP caused by the rare ACVR1^G356D^ mutation, symmetrical glial hyperplasia in the pons and medulla oblongata were identified [112]. In a previous autopsy series, one case of a 31-year-old female with FOP revealed a gliotic nodule in the floor of the fourth ventricle and additional focal nodularity at the lateral recess of the fourth ventricle at the ponto-medullary junction [40]. Additionally, another case report of a child with FOP presenting with hydrocephalus described symmetrical swelling and T2 hyperintensity of the dentate nuclei, a T2 hyperintensity in dorsal pons, and a non-enhancing T2 hyperintense soft tissue mass around the ventral pons invading the lateral recesses of the fourth ventricle [113]. Further in-depth studies on patients with neurologic symptoms and correlation with imaging findings are needed to better understand the neurologic effects of ACVR1 overactivity.

### 3.5. ACVR1 and Diffuse Intrinsic Pontine Glioma (DIPG)

Diffuse Intrinsic Pontine Glioma (DIPG) is a fatal pediatric brainstem tumor. Outcomes for patients are universally dismal with a median survival time of only 9–12 months [114]. Effective treatments are limited as the anatomical location of the tumor makes surgery difficult to perform, and there is a lack of efficacious chemotherapies [115]. In four independent studies using whole genome sequencing of patients with DIPG, recurring, activating heterozygous somatic mutations in ACVR1 were found in about 25% of patients with DIPG [115,116,117,118]. Of the seven ACVR1-associated mutations found [119], six of them have been described in patients with FOP including R206H, Q207E, R258G, G328E, G328W, and G356D. The seventh mutation G328V is only observed in DIPG at a frequency of 28%. R206H mutation is the second most common ACVR1 mutation at a frequency of 20% in DIPG [119]. Unlike FOP, DIPG has additional mutations co-occurring with ACVR1 mutations. For example, the most common mutation observed in 80% of patients with DIPG is a K27M substitution occurring more commonly in histone 3 encoding genes (H3^K27M^) [116,117,120,121]. About 55% of DIPG also harbor mutations in *Pik2ca* that hyperactivate phosphoinositide-3-kinase signaling [122].

H3^K27M^ and ACVR1 mutations occur very early during tumorigenesis in DIPG patients [123]. Compared to those carrying wild-type ACVR1, patients with DIPG that carried a mutant copy of ACVR1 were found to have a younger age at diagnosis, longer survival, and were more commonly females [115,118]. In preclinical animal models, in combination with platelet-derived growth factor A (PDGFA) signaling, Hoeman et al. recently reported that ACVR1^R206H^ and H3.1^K27M^ significantly increased tumor incidence and decreased mouse survival [124]. Conversely, inhibiting the ACVR1 signaling decreased the cell proliferation in vitro and prolonged mouse survival in vivo. In another study [125], ACVR1^G328V^ caused the arrest of the oligodendroglial differentiation and induced gliomas in mice in cooperation with H3b^K27M^ and Pik3ca^H1047R^ [125]. To date, although the dysregulated ACVR1 signaling may contribute to neurological dysfunction, there is no clinical evidence that patients with FOP have an increased propensity for the development of CNS malignancies including gliomas.

### 3.6. Nociceptor Regulation of HO

Nociceptors are the free nerve endings of primary sensory neurons. Emerging evidence supports that nociceptors can modulate the host response against tissue injury by releasing neurotransmitters including substance P (SP) and CGRP [126,127,128]. As nociceptor innervation in bone is well known [129,130], a number of human and animal studies implicate neuronal control as important regulators in several bone diseases including trauma-induced HO [131,132,133,134,135]. For example, selective denervation of transient receptor potential vanilloid 1 (*Trpv1^+^*) nociceptors induces bone volume loss in rodents [136,137]. On the other hand, HO was dramatically inhibited in *Trpv1*^-/-^ mice [138], implying that the nociceptors and bone formation/repair are closely linked.

BMP2, which can be elevated in vivo during bone and muscle injury [139], was found to induce the release of SP and CGRP from cultured sensory neurons [140]. On the other hand, SP and CGRP can also interact with BMP2 and BMP4 to regulate osteoprogenitor stem cell differentiation in bone formation [141,142,143]. Overexpression of BMP2 in hindlimb muscles with BMP2 adenovirus resulted in HO formation and a significant increase in mast cells within sites of new bone formation [144]. Inhibiting mast cell degranulation by treating mice with cromolyn, significantly reduced BMP2-induced HO compared to the control mice. Moreover, SP can regulate neurological HO by influencing recruitment of immune cells to the injury sites of the CNS [144]. Tuzmen et al. also provided in vivo evidence of the direct contribution of SP to HO formation by injecting SP to murine Achilles tendon [145]. Taken together, nociceptors may be a contributor to HO in FOP, or may serve as possible therapeutic targets to mitigate HO formation in FOP and other related forms of heterotopic ossification.

## 4. Conclusions

In summary, our understanding of the non-skeletal manifestations of FOP is still growing. Awareness of cardiopulmonary and neurologic phenotypes observed in FOP is now increasing. FOP is clearly associated with cardiopulmonary complications such as pulmonary hypertension and thoracic insufficiency syndrome. While the ACVR1 receptor has important roles in cardiac development as revealed by loss of function studies, how increased or neoceptor ACVR1 activity contributes to cardiac defects and conduction abnormalities is still an area of active study. A better understanding of the cardiac phenotypes will help improve future clinical recommendations regarding cardiopulmonary complications. The neurologic phenotype in FOP is notable for neuropathic pain, hyperalgesia, and alterations in CNS imaging. Additional studies are also needed to assess the mechanisms of neuropathic pain, along with other neurological symptoms observed in patients with FOP. Together, these results emphasize that while the bone phenotype is the most dramatic visible consequence, FOP should be considered and managed as a multi-system disease.

## Figures and Tables

**Figure 1 biomedicines-09-00155-f001:**
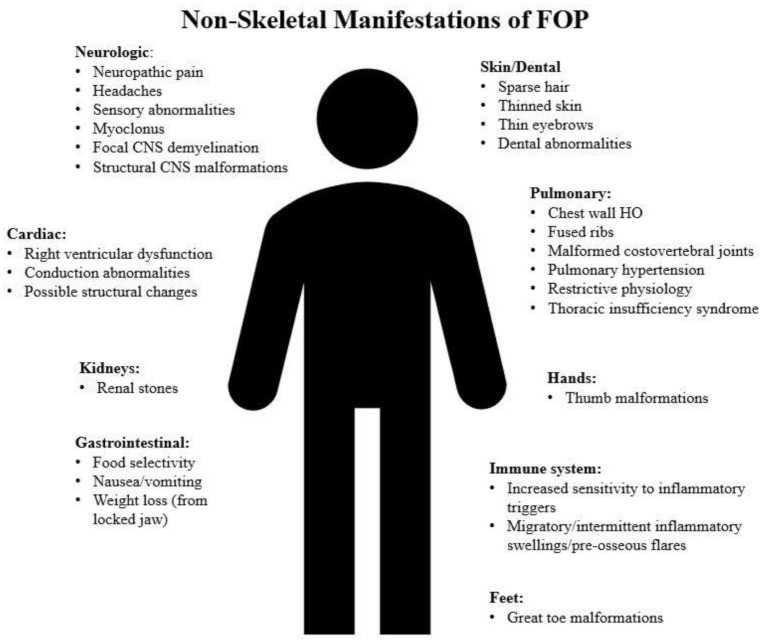
Major non-skeletal manifestations of Fibrodysplasia Ossificans Progressiva (FOP). The hand and foot malformations are also listed, as they are strongly correlated with the diagnosis of FOP. Other skeletal malformations are summarized in Table 1 separately.

**Table 1 biomedicines-09-00155-t001:** Major skeletal anomalies in Fibrodysplasia Ossificans Progressiva (FOP) not caused by heterotopic ossification.

○ Great toe malformations ○ Clinodactyly ○ Malformed thumbs ○ Jaw bone malformations ○ Spinal deformities ○ Osteochondromas especially of proximal medial tibia ○ Short broad femoral necks

## Data Availability

Not Applicable.

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
