# Peer review of "Cardiopulmonary and Neurologic Dysfunctions in Fibrodysplasia Ossificans Progressiva"

_biomedicines, 2021, doi:10.3390/biomedicines9020155_

Round 1

Reviewer 1 Report

This paper addresses the issue of non-skeletal abnormalities in FOP patients, focusing in particular on cardiovascular and neurologic dysfunctions. This is of great interest since, despite the fact that the severe skeletal malformations due to heterotopic ossification is a major cause of disability for the patients, the mutation in almost all tissues and organs may be cause of aggravation of the disease severity and progression. Still these aspects have never been studied in much detail in neither patients nor in animal models.

The article is well written and organized, and provide an extensive literature review and discussion.

Author Response

No changes requested. Thank you for the positive comments!

Reviewer 2 Report

  • This manuscript reviewed the non-skeletal manifestations of the FOP, a classically considered as bone disease. The authors summarized the emerging evidence on cardiopulmonary and neurologic dysfunctions in FOP patients. The authors proposed that FOP should be considered and managed as a multi-system, including bone phenotypes as the most visible one. The citation and contents make sense and broaden scientist's view on FOP as bone diseases only. It will update the colleague's knowledge of the FOP as well.

Author Response

(The authors gave the same response as above.)

Reviewer 3 Report

The authors have carefully listed and summarized the reviews on cardiopulmonary dysfunction in FOP and neurologic dysfunction in FOP, and it is well-written manuscript.

There may be a little more description of the background of FOP using mouse genetics model as an introduction. For example, it would be better if there was a summary of Bmp ligand over expression mouse model (Nse-Bmp4), Alk2Q207D(caAlk2), Acvr1R206H, Acvr1tnR206H knock in mouse model (Shepard et al. Nat commun, 2018). Overall, a good review manuscript.

Author Response

Thank you for this suggestion. Because non-bone phenotypes have not been studied in great detail in FOP mouse models (no published studies to date on cardiac effects of the FOP mutations, and only limited data on neurologic phenotpyes), we decided to focus the background information of FOP on the patient clinical phenotypes. We added new text throughout the review (e.g., Page 3 Line 89) to introduce this limitation and also discuss any relevant non-bone phenotypes in the major published mouse models.

Reviewer 4 Report

The manuscript entitled “Cardiopulmonary and Neurologic dysfunctions in Fibrodysplasia Ossificans Progressiva” by Fatima Khan, Xiaobing Yu and Edward C. Hsiao describes non-skeletal manifestations of FOP, in particular cardiopulmonary and neurologic dysfunctions.

It is an interesting point of view, not only to provide an extended picture of FOP clinical manifestations and their consequences for the affected patients, but also to add elements of discussion on the multiple consequences of the ACVR1 mutation causative of FOP in cell types and tissues where the BMP signaling pathways play a role in development and in postnatal life.

Regarding cardiac anomalies, there are 2 references published by the same group, reporting ACVR1 mutations/variants (Irene C Joziasse et al. ALK2 mutation in a patient with Down’s syndrome and a congenital heart defect. European Journal of Human Genetics (2011) 19, 389–393; Kelly A. Smith et al. Dominant-Negative ALK2 Allele Associates With Congenital Heart Defects. Circulation. 2009;119:3062-3069) that have been puzzling the FOP community. My suggestion to the authors is to comment those papers.

In page 4 lines 123,124 the authors make an appropriate statement. I suggest they extend this statement considering that any injury condition that takes place in the heart, but in general also in other organs/tissues, can cause the release of activin A which, in FOP patients, results in further upregulation of BMP signaling. As an example this reference: Arne Yndestad et al. Elevated Levels of Activin A in Heart Failure. Potential Role in Myocardial Remodeling. Circulation. 2004;109:1379-1385. doi: 10.1161/01.cir.0000120704.97934.41.

Following this observation, also for the nervous system many pathological conditions can induce Activin A production and release with consequences due to BMP signaling overactivation. The authors might consider several reports on this issue.

Another suggestion is to cite and comment a very recent paper: Mori S, Suzuki SO, Honda H, Hamasaki H, Sakae N, Sasagasako N, Furuya H, Iwaki T. Symmetrical glial hyperplasia in the brainstem of fibrodysplasia ossificans progressiva. Neuropathology. 2021 Jan 6. doi: 10.1111/neup.12715.

Minor

page 5 line 209: is it embryonical or perinatal lethality?

Author Response

We are delighted by the positive responses from the reviewers, and greatly appreciate their suggestions. We have incorporated those suggestions to strengthen our paper.

1) Regarding cardiac anomalies, there are 2 references published by the same group, reporting ACVR1 mutations/variants (Irene C Joziasse et al. ALK2 mutation in a patient with Down’s syndrome and a congenital heart defect. European Journal of Human Genetics (2011) 19, 389–393; Kelly A. Smith et al. Dominant-Negative ALK2 Allele Associates With Congenital Heart Defects. Circulation. 2009;119:3062-3069) that have been puzzling the FOP community. My suggestion to the authors is to comment those papers.

Thank you for these suggestions. These papers and a brief discussion of their relevance have been added to our review (Paragraph starting Page 4, line 143).

2) In page 4 lines 123,124 the authors make an appropriate statement. I suggest they extend this statement considering that any injury condition that takes place in the heart, but in general also in other organs/tissues, can cause the release of activin A which, in FOP patients, results in further upregulation of BMP signaling. As an example this reference: Arne Yndestad et al. Elevated Levels of Activin A in Heart Failure. Potential Role in Myocardial Remodeling. Circulation. 2004;109:1379-1385. doi: 10.1161/01.cir.0000120704.97934.41.

Thank you for this suggestion. We added this reference and some additional discussion to the cardiac sections (broadly in section 2.2, and specifically on page 4, lines 187-190). One intriguing aspect is that while these and other studies suggest that different forms of tissue injury can cause Activin A production, a specific biological effect may or may not manifest. For example, while cardiac muscle can also be injured in patients with FOP, we have seen no cases of cardiac muscle calcification. We try to balance this discussion with the importance of organ-specific responses, which may not have been identified yet.

3) Following this observation, also for the nervous system many pathological conditions can induce Activin A production and release with consequences due to BMP signaling overactivation. The authors might consider several reports on this issue.

Thank you for this suggestion. We added several references and discussion to the neurology sections (sections 3.2-3.4, with specific reference to Activin A on Page 7 line 393-396)

4) Another suggestion is to cite and comment a very recent paper: Mori S, Suzuki SO, Honda H, Hamasaki H, Sakae N, Sasagasako N, Furuya H, Iwaki T. Symmetrical glial hyperplasia in the brainstem of fibrodysplasia ossificans progressiva. Neuropathology. 2021 Jan 6. doi: 10.1111/neup.12715.

We were also very excited to see this recent paper. This has now been added to our manuscript (Page 8 line 452).

5) Page 5 line 209: is it embryonical or perinatal lethality?

Embryonic lethality is observed in homozygous Alk2 mutant mice [Mishina, 1999, Page 318].